# Long-Term Sequential Digital Dermoscopy of Low-Risk Patients May Not Improve Early Diagnosis of Melanoma Compared to Periodical Handheld Dermoscopy

**DOI:** 10.3390/cancers15041129

**Published:** 2023-02-10

**Authors:** Riccardo G. Borroni, Vincenzo Panasiti, Mario Valenti, Luigi Gargiulo, Giuseppe Perrone, Roberta Dall’Alba, Clarissa Fava, Francesco Sacrini, Luca L. Mancini, Sofia A. A. M. Manara, Emanuela Morenghi, Antonio Costanzo

**Affiliations:** 1Department of Biomedical Sciences, Humanitas University, Via Rita Levi Montalcini, 4, 20072 Pieve Emanuele, Italy; 2Dermatology Unit, Humanitas Research Hospital—IRCCS, Via Alessandro Manzoni, 56, 20089 Rozzano, Italy; 3Department of Plastic Surgery and Dermatology, Università Campus Bio-Medico di Roma, Via Alvaro del Portillo, 21, 00128 Rome, Italy; 4Fondazione Policlinico Universitario Campus Bio-Medico, Via Alvaro del Portillo, 200, 00128 Rome, Italy; 5Department of Pathology, Università Campus Bio-Medico di Roma, Via Alvaro del Portillo, 21, 00128 Rome, Italy; 6Pathology Unit, Humanitas Research Hospital—IRCCS, Via Alessandro Manzoni, 56, 20089 Rozzano, Italy; 7Biostatistics Unit, Humanitas Research Hospital—IRCCS, Via Alessandro Manzoni, 56, 20089 Rozzano, Italy

**Keywords:** melanoma, dermoscopy, epiluminescence microscopy, sequential digital dermoscopy

## Abstract

**Simple Summary:**

The combined use of total-body photography and sequential digital dermoscopy has been validated as a standard of care for the early detection of cutaneous melanoma in high-risk individuals. However, the exact impact of this approach in low-risk subjects is not clear. We conducted a retrospective study to evaluate whether there were any differences of prognostic significance between melanomas diagnosed with handheld dermoscopy and sequential digital dermoscopy. In our study, no significant differences were detected in terms of tumor thickness and stage distribution in low-risk patients followed in the long term. Our data contribute to shape the role of sequential digital dermoscopy in patients without risk factors for melanoma.

**Abstract:**

Sequential digital dermoscopy (SDD) enables the diagnosis of a subgroup of slow-growing melanomas that lack suspicious features at baseline examination but exhibit detectable change on follow-up. The combined use of total-body photography and SDD is recommended in high-risk subjects by current guidelines. To establish the usefulness of SDD for low-risk individuals, we conducted a retrospective study using electronic medical records of low-risk patients with a histopathological diagnosis of cutaneous melanoma between 1 January 2016 and 31 December 2019, who had been referred and monitored for long-term follow-up of clinically suspicious melanocytic nevi. We sought to compare the distribution of “early” cutaneous melanoma, defined as melanoma in situ and pT1a melanoma, between SDD and periodical handheld dermoscopy in low-risk patients. A total of 621 melanomas were diagnosed in a four-year timespan; 471 melanomas were diagnosed by handheld dermoscopy and 150 by digital dermoscopy. Breslow tumor thickness was significantly higher for melanomas diagnosed by handheld compared to digital dermoscopy (0.56 ± 1.53 vs. 0.26 ± 0.84, *p* = 0.030, with a significantly different distribution of pT stages between the two dermoscopic techniques. However, no significant difference was found with respect to the distribution of pT stages, mean Breslow tumor thickness, ulceration, and prevalence of associated melanocytic nevus in tumors diagnosed on periodical handheld dermoscopy compared to SDD. Our results confirm that periodical dermoscopic examination enables the diagnosis of cutaneous melanoma at an earlier stage compared to first-time examination as this was associated in our patients with better prognostic features. However, in our long-term monitoring of low-risk subjects, Breslow tumor thickness and pT stage distribution did not differ between handheld periodical dermoscopy and SDD.

## 1. Introduction

Melanoma is one of the most aggressive skin cancers, being responsible for the largest percentage of deaths among cutaneous neoplasms [1]. Even with the availability of more effective treatments for advanced and metastatic cutaneous melanoma, early diagnosis is still crucial to achieve complete cure or to prolong overall survival. Historically, screening for melanoma has been carried out by naked-eye examination of pigmented lesions of the skin, which is nowadays still performed by both lay people and general practitioners, using the “ABCDE” rule (presence of asymmetry, irregular borders, multiple and/or inhomogeneous colors, diameter > 6 mm, and rapid evolution) or the “ugly duckling” rule (standout lesion which is very different from all the other lesions of the patient) [2]. Dermoscopy (dermatoscopy, epiluminescence microscopy) was first introduced in the late 1980s of the 20th century and is currently employed in daily clinical practice by virtually any dermatologist worldwide [3]. A handheld dermoscope (or dermatoscope) is a device equipped with a magnification lens and a light source, which enables the non-invasive visualization of the subsurface morphology of cutaneous lesions, down to the depth of the superficial dermis. It reveals colors and structures that are normally not visible to the unaided eye, correlating with histopathological features [4]. When performed by a trained physician, dermoscopy is more accurate than naked-eye examination for the diagnosis of cutaneous melanoma [5,6,7] for which it represents the current gold standard for early detection [1]. According to a meta-analysis conducted on 104 study publications, dermoscopy can increase the sensitivity for the detection of melanoma over the naked-eye total-body examination from 76% to 92% [8]. Today, dermoscopy must be considered as a first-level screening tool to increase the rate of melanoma excised at an early stage. Conventional handheld dermoscopy today is performed using inexpensive, portable, and fast tools [9]. For all these reasons, handheld dermoscopy has become part of routine clinical inspection of pigmented skin lesions. However, in some cases malignant melanoma may be difficult to distinguish from benign melanocytic nevi also by dermoscopy, particularly in incipient lesions in which specific dermoscopic criteria for melanoma may not be evident [10]. In such melanocytic lesions, the history of morphologic changes may be a clue to the correct diagnosis of melanoma [11,12]. Sequential digital dermoscopy (SDD), i.e., the capture, storage, and comparison of successive dermoscopic images, separated by an interval of time, of one or multiple melanocytic lesions, aids in the detection of suspicious change, enabling the diagnosis of a subgroup of slow-growing melanomas that lack suspicious features at baseline examination, but exhibit detectable changes on follow-up [13]. Total-body photography enables the documentation of lesions and the early detection of any concerning changes over time. Digital photographs of skin surface areas standardized based on body positions and digital dermoscopic images of lesions of concern allow for nearly the entire body surface to be recorded and referred to in follow-up examinations and re-photographed for comparison at the follow-up visit. Rhodes proposed that baseline photography and consecutive removal of suspicious new or changing nevi could reduce costs and increase reassurance in individuals at high risk of developing melanoma, compared to wholesale surgical excision of all nevi in this group of subjects [14]. The combined use of total-body photography and SDD, called “two-step method of digital follow-up”, has been proposed [15] and is recommended in high-risk subjects by current guidelines [16]. It has been shown that, using digital dermoscopy, the percentage of in situ melanomas and thin melanomas is higher than expected in the general population [17]. The probability of detecting a melanoma during digital monitoring increases with the duration of follow-up [18]. On the other hand, patients without risk factors for melanoma are often referred to our clinic by general practitioners or themselves with a generic request of total body photography and/or SDD for otherwise overtly bland melanocytic nevi.

To establish the usefulness of SDD for low-risk individuals, we studied the distribution of “early” cutaneous melanoma, defined as melanoma in situ and pT1a melanoma, in low-risk patients diagnosed by SDD as compared to periodical handheld dermoscopy. In addition, we assessed whether this distribution was different between those who were diagnosed with melanoma after only one dermatological examination or after repeated (≥2) periodical examinations.

## 2. Patients and Methods

We conducted a retrospective study using electronic medical records of low-risk patients with a histopathological diagnosis of cutaneous melanoma between 1 January 2016 and 31 December 2019, who had been referred and monitored at our Dermatology Units for long-term follow-up of clinically suspicious melanocytic nevi which did not show any dermoscopic feature of melanoma on first examination. Patients who were diagnosed with melanoma during the same time frame but had a family and/or personal history of cutaneous melanoma, those who had a melanoma excised but had not been examined previously at our dermatology centers, and those with incomplete or missing data were excluded. Low-risk patients were defined as those without family or personal history of cutaneous melanoma and with nevus count < 50 [18]. We also excluded patients with personal history of non-melanoma skin cancers, immunosuppression (including immune-suppressive treatments or concomitant immunodeficiencies), or multiple courses of phototherapy. Patients were included if, before the histopathological diagnosis of melanoma, they had undergone at least one dermoscopic examination at our clinics, either handheld or digital. Patients were assigned the periodic examination group only if they underwent SDD or periodical handheld dermoscopy with a long-term (6–12-month) interval [19]. Patients diagnosed with melanoma on handheld dermoscopy were considered in the group of handheld dermoscopy even if they had undergone long-term SDD in the past, because at the time of latest examination the objective comparison of digital dermoscopic images could not be demonstrated. At least two different dermatologists trained in dermoscopy independently assessed the patients by either handheld dermoscopy or digital dermoscopy in each clinic. FotoFinder Medicam 800HD Dermoscope (FotoFinder Systems GmbH, Bad Birnbach, Germany) was used for digital videodermoscopy while Heine Delta 20T (HEINE Optotechnik GmbH & Co. KG, Gilching, Germany) was used for handheld dermoscopy at both Dermatology Units.

For each patient, demographic data (including age, gender and anatomical site affected), the technique of dermoscopic examination (handheld or digital), the number of dermoscopic examinations, follow-up interval, and duration of dermoscopic follow-up were recorded. The 7-point checklist score based on dermoscopic pattern analysis [20] was employed for the evaluation of dermoscopic features of each lesion. Presence of associated melanocytic nevus and pTNM stage according to the American Joint Committee on Cancer (AJCC) 8th edition [21] were retrieved from the histopathological report as well as Breslow tumor thickness and ulceration in case of invasive melanomas. American Joint Committee on Cancer 8th edition staging [21] was used for all melanomas including those who had been diagnosed under the previous edition at the time of original diagnosis. Institutional review board approval was exempted as the retrospective study protocol no. 43/22 did not deviate from standard clinical practice. All included patients had provided written consent for retrospective study of data collected during routine clinical practice (including both demographics and clinical data). The study was performed in accordance with the Helsinki Declaration of 1964 and its later amendments. Data collection and handling complied with applicable laws, regulations, and guidance regarding patient protection, including patient privacy.

### Statistical Analysis

Due to the exploratory nature of this study, its overarching goal was to assess the performance of SDD and handheld dermoscopy in time. Since no significant difference between the two techniques was anticipated, all the eligible patients were included in a 4-year time frame, without a sample size calculated a priori. Discrete parameters were reported as count and percentage. Continuous parameters were reported using frequency, mean, and standard deviation (SD) values. The differences between the different techniques were explored with chi-squared test or Mann–Whitney test, as appropriate. A *p* value < 0.05 was considered statistically significant.

## 3. Results

A total of six hundred and twenty-one cutaneous melanomas were diagnosed between 1 January 2016 and 31 December 2019 in 606 consecutive low-risk patients (301 females, 305 males; mean age 54.56 ± 15.85 years). Demographic, clinical, and histopathological data of all 621 melanomas are summarized in Table 1 and Table 2. A total of 471 melanomas were diagnosed by handheld dermoscopy and 150 by digital dermoscopy. Melanoma was diagnosed at a mean older age on handheld dermoscopy compared to digital dermoscopy (56.65 ± 15.92 years and 47.99 ± 13.74 years, respectively; *p* < 0.001; Table 1). Breslow tumor thickness was significantly higher for melanomas diagnosed by handheld compared to digital dermoscopy (0.59 ± 1.61 mm and 0.21± 0.36 mm, respectively; *p* = 0.004, Table 1).

A total of four hundred and eighty-seven melanomas (78.4%) were identified on first-time dermatological examination, although the proportion was statistically different between those diagnosed by handheld or digital dermoscopy (84.08% and 60.67%, respectively; *p* < 0.001). Mean age at diagnosis on first-time dermoscopy and periodical dermoscopy were comparable (54.9 ± 15.95 vs. 53.32 ± 15.51, *p* = 0.307, Table 2). Melanomas diagnosed on first-time dermoscopy overall exhibited significantly higher mean Breslow tumor thickness compared to periodical dermoscopy (0.56 ± 1.53 vs. 0.26 ± 0.84, *p* = 0.030, Table 2). Consistent with these data was the finding of higher, although not significant, Breslow thickness in melanomas diagnosed on first-time handheld dermoscopy compared to periodical handheld dermoscopy (mean Breslow thickness 0.63 ± 1.68 mm vs. 0.36 ± 1.10 mm; *p* = 0.079; Table 1; Figure 1). On the other hand, Breslow thickness was comparable between first-time digital dermoscopy and SDD (0.52 ± 0.48 mm vs. 0.42 ± 0.17 mm; *p* = 0.194 Table 1, Figure 1).

Melanomas diagnosed on first-time handheld dermoscopy had a significant higher mean Breslow tumor thickness compared to those detected on first-time digital dermoscopy (0.63 ± 1.68 mm and 0.25 ± 0.42 mm, respectively; *p* < 0.001; Table 2; Figure 2) and presented in significantly older patients (56.48 ± 16.32 years and 48.97 ± 14.53 years, respectively; *p* < 0.001). Remarkably, no significant difference was found with respect to the distribution of pT stages, mean Breslow tumor thickness (0.36 ± 1.10 mm and 0.13 ± 0.22 mm, respectively; *p* = 0.092), ulceration, and prevalence of associated melanocytic nevus in tumors diagnosed on periodical handheld dermoscopy compared to SDD (Figure 3, Table 2). Mean time to diagnosis during periodical follow-up examinations were not significantly different between periodical handheld dermoscopy and SDD (40.93 ± 37.04 months and 31.33 ± 22.54 months, respectively; *p* = 0.067) as were the mean number of examinations to diagnosis (3.08 ± 1.54 and 3.05 ± 0.7, respectively; *p* = 0.910) and the mean follow-up period (24.25 ± 24.63 months for handheld dermoscopy and 17.23 ± 15.66 months for SSD; *p* = 0.056) (Table 2).

Lastly, we focused on the clinical and dermoscopic characteristics of melanomas detected with SDD. A total of fifty-nine melanomas were detected during a digital dermoscopy follow-up examination. A total of 49 out of 59 lesions (83.1%) were excised because they displayed changes since the last digital dermoscopy. The main reason for the surgical excision of these lesions was the evidence of concomitant increase in size and pattern change since the latest digital dermoscopy examination (in 19 melanomas). A total of 15 lesions exhibited an increase in size only and the other 15 exhibited change in dermoscopic pattern only. A total of 10 out of 59 lesions (16.9%) were excised after SDD because they were not evident on previous total-body photography. No significant difference in the 7-point checklist score based on patter analysis was observed between melanomas identified on first-time digital dermoscopy and those detected during SDD. Likewise, there was no significant difference in the 7-point checklist score between the 10 melanomas detected on SDD, which were not evident on previous digital dermoscopy, and the 91 melanomas detected on first-time digital dermoscopy.

## 4. Discussion

Overall, our results confirm that periodical dermoscopy carried out by a trained physician, regardless of the technique used, enables the diagnosis of cutaneous melanoma at an earlier stage compared to first-time examination [22]. In our patients, this was associated with better prognostic features such as a lower Breslow thickness. Besides its retrospective approach, one of the limitations of our study is the relatively low number of melanomas diagnosed by digital dermoscopy (and by SDD, in particular), with even fewer invasive melanomas within this group. In our setting, the overall higher number of melanomas diagnosed on first-time examination compared to periodical dermoscopic examination, irrespective of the technique, could be explained by the more common referral of low-risk patients to first-level, handheld dermoscopy of suspicious lesions by the general practitioner or the patients themselves, as opposed to direct, self-referred access to a second-level examination including digital storage of dermoscopic images of lesions with some likelihood of subsequent surgical removal. Similarly, the significantly higher Breslow thickness in patients diagnosed by first-time handheld dermoscopy compared to first-time digital dermoscopy could be explained by a different mode of referral. Low-risk patients presenting with a changing or suspicious lesion usually access handheld dermoscopy, since possible subsequent surgical removal may not necessarily imply digital image capture for further follow-up of that lesion [23]. Conversely, a proportion of low-risk patients (14.65% in our study) happen to be diagnosed with melanoma on first-time digital dermoscopy, possibly at the beginning of a secondary prevention schedule that may later imply SDD. In our study, the majority (60.67%) of melanomas diagnosed by digital dermoscopy was in fact detected on first-time examination, indicating that in these low-risk patients, the capture and storage of digital dermoscopic images for further SDD would not have been necessary a priori. This is further supported by similar Breslow thicknesses of invasive melanomas and pT stage distribution in first-time digital dermoscopy and SDD. Moreover, we did not observe a significant difference in 7-point checklist score between the 10 melanomas detected on SDD, which were not evident on previous digital dermoscopy, and the melanomas detected on first-time digital dermoscopy, showing that a sequential approach would have not be needed in those patients, as those melanomas would have been diagnosed at the first examination based on dermoscopic criteria.

Together with younger age at diagnosis of patients undergoing digital dermoscopy, our data suggest that digital dermoscopy could select low-risk individuals with higher socio-economic status and better prognosis [24,25,26]. These findings could be explained by the fact that digital dermoscopy is generally not reimbursed by the Italian national healthcare system to all low-risk patients but can be funded by private insurances or the patients themselves in our specific healthcare setting. Our findings may have relevant implications for routine clinical practice. Although Kofler et al. estimated that the time required to perform a single SDD examination is 7 min (about 15 s for lesion) [27], in our experience, this may be true only for SDD examinations subsequent to the first one, once family and personal history had already been collected, the entire skin surface of the patient had been photographed according to standardized poses, and each suspicion lesion to be mapped had been marked and baseline dermoscopic photographs had been acquired for each lesion to be subsequently monitored on follow-up. On the other hand, it has also been shown that handheld dermoscopy is less time-consuming. A study conducted by Zalaudek et al. [28] randomly divided 1328 patients with at least one cutaneous lesion to undergo a complete dermatological visit, with or without dermoscopy, to determine the time required to perform each examination. According to this study, the median time needed for complete dermatological visit with handheld dermoscopy was 142 s, less than 3 min. The combination of total body photography and SDD is indeed more time-consuming compared to handheld dermoscopy. This may be even more compelling considering the proportion of melanomas that were excised after at least two SDD examinations, resulting in overall increased examination time for SDD. Patient’s compliance may also represent a critical issue. In relationship with the frequency of dermatological examinations, regardless of the dermoscopic technique used, the importance of patient’s compliance to follow-up was demonstrated by Kittler and Binder [29], who concluded that SSD is a useful procedure in patients with multiple atypical nevi but that should be use judiciously in terms of case selection and dermatologist’s expertise. Argenziano et al. [30] observed that patient’s compliance is related to the follow-up protocol, suggesting that a short term follow-up, with a 3 months first re-examination, makes it possible to reach an 84% rate of compliance in the cohort of patients they have examined.

Although the calculation of the number of digital dermoscopic examination needed to excise one melanoma is beyond the scope of this study, the relatively low number of melanomas diagnosed by digital dermoscopy compared to handheld dermoscopy in our 4-year study seem to be in line with the findings of Schiffner and colleagues, who after a median follow up of 24 months did not detect any melanoma by SDD in 145 low-risk individuals, concluding that long-term SDD seems to not be helpful for the detection of melanoma in these subjects [18].

## 5. Conclusions

Our results confirm that regular dermoscopic examinations allow for earlier diagnosis of cutaneous melanoma compared with initial examination, as this was associated with better prognostic features in our patients regardless of the dermoscopy technique used. However, in our long-term monitoring of low-risk subjects, Breslow tumor thickness and pT stage distribution did not differ between handheld periodical dermoscopy and SDD, suggesting that the precise selection of patient groups and of the lesions to be monitored is fundamental for an effective SDD that also achieves a good cost/efforts ratio [31]. Although limited, our findings emphasize the clinical significance of directing low-risk individuals to the most appropriate dermoscopic technique, with the aim of optimizing resources on one hand and reassuring the patient for otherwise bland melanocytic nevi that could be erroneously perceived as potentially at risk if sampled for subsequent digital dermoscopic follow-up. Additional studies, with larger cohorts of patients and possibly with a prospective design, are strongly needed to further validate the usefulness of long-term SDD in the general population.

## Figures and Tables

**Figure 1 cancers-15-01129-f001:**
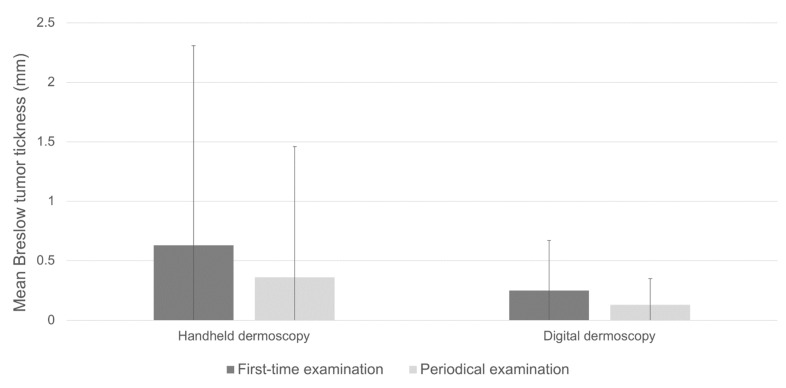
Breslow thickness in invasive melanomas diagnosed by handheld dermoscopy compared to digital dermoscopy.

**Figure 2 cancers-15-01129-f002:**
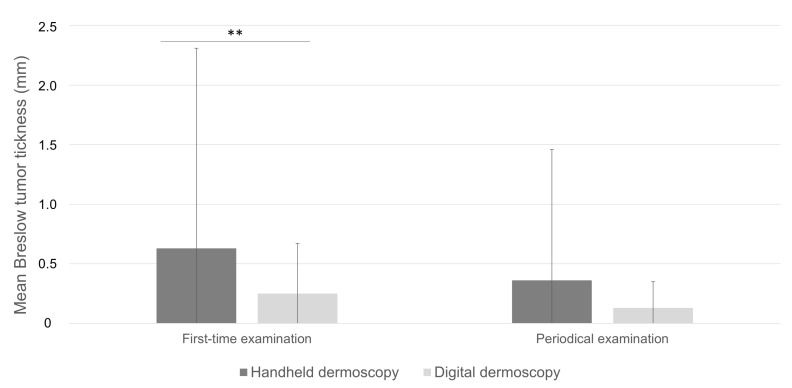
Mean Breslow tumor thickness in melanomas diagnosed on first-time examination compared to periodical dermoscopy. **: *p* < 0.01.

**Figure 3 cancers-15-01129-f003:**
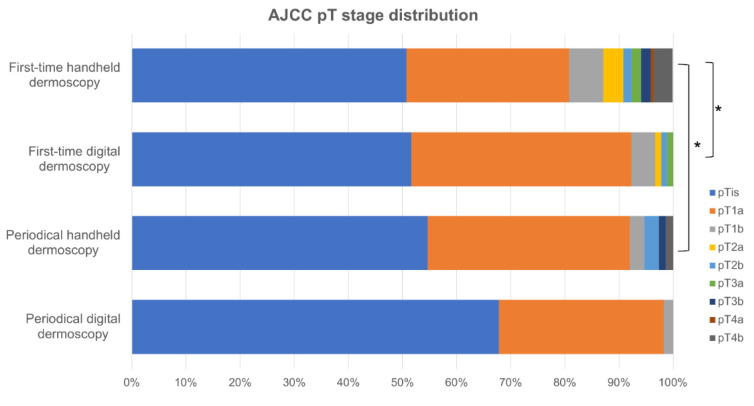
Distribution of pT stages between the groups. * *p* < 0.05.

**Table 1 cancers-15-01129-t001:** Demographic, clinical, and histopathological data of melanomas diagnosed by handheld and digital dermoscopy. ns, non-significant.

	Handheld Dermoscopy				Digital Dermoscopy			
	All (*n =* 471)	First-Time (*n* = 396)	Periodical (*n* = 75)	*p*	All (*n =* 150)	First-Time (*n* = 91)	Sequential (*n* = 59)	*p*
Age (years)	56.65 ± 15.92	56.26 ± 15.96	58.69 ± 15.67	ns	47.99 ± 13.74	48.97 ± 14.53	44.49 ± 12.38	ns
Gender								
Females	233 (49.47%)	198 (50%)	40 (53.33%)	<0.01	75 (50%)	46 (50.55%)	29 (47.54%)	ns
Males	238 (50.53%)	198 (50%)	35 (46.67%)		75 (50%)	45 (49.45%)	30 (52.46%)	
Location								
Head/neck	33 (7.01%)	28 (7.07%)	5 (6.67%)	<0.01	6 (4%)	1 (1.1%)	5 (8.47%)	<0.05
Trunk	252 (53.50%)	212 (53.54%)	40 (53.33%)		93 (62%)	34 (60.44%)	38 (64.41%)	
Upper limb	69 (14.65%)	59 (14.9%)	10 (1.33%)		11 (7.33%)	8 (8.79%)	3 (5.08%)	
Lower limb	117 (24.84%)	97 (24.49%)	20 (26.67%)		40 (26.67%)	27 (29.67%)	13 (22.03%)	
Associated nevus	150 (31.85%)	125 (31.56%)	25 (33.33%)	ns	63 (42%)	38 (41.76%)	25 (42.37%)	ns
AJCC pT stage								
pTis	242 (51.38%)	201 (50.76%)	41 (54.66%)	<0.05	87 (58%)	47 (51.65%)	40 (67.8%)	ns
pT1a	147 (31.21%)	119 (30.05%)	28 (37.33%)		55 (36.67%)	37 (40.66%)	18 (30.51%)	
pT1b	27 (5.73%)	25 (6.31%)	2 (2.67%)		5 (3.33%)	4 (4.4%)	1 (1.7%)	
pT2a	15 (3.18%)	15 (3.70%)	0		1 (0.67%)	1 (1.1%)	0	
pT2b	8 (1.7%)	6 (1.51%)	2 (2.67%)		1 (0.67%)	1 (1.1%)	0	
pT3a	7 (1.49%)	7 (1.77%)	0		1 (0.67%)	1 (1.1%)	0	
pT3b	8 (1.7%)	7 (1.77%)	1 (1.33%)		0	0	0	
pT4a	2 (0.42%)	2 (0.51%)	0		0	0	0	
pT4b	15 (3.18%)	14 (3.53%)	1 (1.33%)		0	0	0	
Breslow thickness (mm)	0.59 ± 1.61	0.63 ± 1.68	0.36 ± 1.10	ns	0.21± 0.36	0.25 ± 0.42	0.13 ± 0.22	ns
Ulceration	42 (8.9%)	37 (9.34%)	5 (6.67%)	<0.01	2 (1.33%)	1 (1.1%)	1 (1.69%)	ns
Time to diagnosis (months)	6.77 ± 21.39	-	40.93 ± 37.04	-	12.32 ± 20.82	-	31.33 ± 22.54	-
Number of examinations	1.34 ± 0.98	1	3.08 ± 1.54	-	1.81 ± 1.35	1	3.05 ± 1.44	-
Follow-up interval (months)	4.01 ± 13.08	-	24.25 ± 24.63	-	6.78 ± 12.91	-	17.23 ± 15.66	-

**Table 2 cancers-15-01129-t002:** Demographic, clinical, and histopathological data of melanomas diagnosed by first-time and periodical dermoscopy. ns, non-significant.

	First-Time Dermoscopy				Long-Term Periodical Dermoscopy			
	All (*n =* 487)	Handheld (*n =* 396)	Digital (*n* = 91)	*p*	All (*n =* 134)	Handheld (*n* = 75)	Digital (SDD) (*n* = 59)	*p*
Age (years)	54.9 ± 15.95	56.26 ± 15.96	48.97 ± 14.53	<0.01	53.32 ± 15.51	58.69 ± 15.67	46.49 ± 12.38	<0.01
Gender					64 (47.76%)	35 (46.67%)	29 (49.15%)	
Females	244 (50.1%)	198 (50%)	46 (50.55%)	ns				ns
Males	243 (49.9%)	198 (50%)	45 (49.45%)		70 (52.24%)	40 (53.33%)	30 (50.85%)	
Location								
Head/neck	29 (5.95%)	28 (7.07%)	1 (1.1%)	<0.01	10 (7.46%)	5 (6.67%)	5 (6.67%)	ns
Trunk	267 (54.83%)	212 (53.54%)	34 (60.44%)		78 (58.21%)	40 (53.33%)	38 (50.67%)	
Upper limb	67 (13.76%)	59 (14.9%)	8 (8.79%)		13 (9.7%)	12 (16%)	1 (1.69%)	
Lower limb	124 (25.46%)	97 (24.49%)	27 (29.67%)		33 (24.63%)	20 (26.67%)	13 (22.03%)	
Associated nevus	163 (33.47%)	125 (31.56%)	38 (41.76%)	<0.01	50 (37.31%)	25 (33.33%)	25 (42.37%)	ns
AJCC pT stage								
pTis	248 (50.92%)	201 (50.76%)	47 (51.65%)	<0.05	81 (60.45%)	41 (54.67%)	40 (67.8%)	ns
pT1a	156 (32.03%)	119 (30.05%)	37 (40.66%)		46 (34.33%)	28 (37.33%)	18 (30.51%)	
pT1b	29 (5.95%)	25 (6.31%)	4 (4.4%)		3 (2.24%)	2 (2.67%)	1 (1.69%)	
pT2a	16 (3.29%)	15 (3.70%)	1 (1.1%)		0	0	0	
pT2b	7 (1.44%)	6 (1.51%)	1 (1.1%)		2 (1.49%)	2 (2.67%)	0	
pT3a	8 (1.64%)	7 (1.77%)	1 (1.1%)		0	0	0	
pT3b	7 (1.44%)	7 (1.77%)	0		1 (0.75%)	1 (1.33%)	0	
pT4a	2 (0.41%)	2 (0.51%)	0		0	0	0	
pT4b	14 (2.87%)	14 (3.53%)	0		1 (0.75%)	1 (1.33%)	0	
Breslow thickness (mm)	0.56 ± 1.53	0.63 ± 1.68	0.25 ± 0.42	<0.01	0.26 ± 0.84	0.36 ± 1.1	0.13 ± 0.22	ns
Ulceration	38 (7.8%)	37 (9.34%)	1 (1.1%)	<0.01	6 (4.48%)	5 (6.67%)	1 (1.69%)	ns
Time to diagnosis (months)	-	-	-	-	36.7 ± 31.74	40.93 ± 37.04	31.33 ± 22.54	ns
Number of examinations	1	1	1	-	3.07 ± 1.49	3.08 ± 1.54	3.05 ± 1.44	ns
Follow-up period (months)	-	-	-	-	21.16 ± 21.37	24.25 ± 24.63	17.23 ± 15.66	ns

## Data Availability

Data supporting the results of this research are available at https://doi.org/10.5281/zenodo.7566534. Accessed on 12 January 2023.

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
