# Peer review of "Long-Term Sequential Digital Dermoscopy of Low-Risk Patients May Not Improve Early Diagnosis of Melanoma Compared to Periodical Handheld Dermoscopy"

_cancers, 2023, doi:10.3390/cancers15041129_

Round 1
Reviewer 1 Report
The manuscript entitled “Long-term sequential digital dermoscopy of low-risk patients may not improve early diagnosis of melanoma compared to periodical handheld dermoscopy” by Riccardo G. Borroni et al., was presented as an original Article in which the authors discussed about the potential benefits and disadvantages in the usage of SDD or handheld dermoscopy for the diagnosis of melanoma. The discussion on their findings seems not to carry full conviction. Difference in sample size (n Handheld dermoscopy = 471 versus n Digital dermoscopy = 150) represents the major limitation of this study. There are chances of detect or failing to detect the significance difference even when it is present. The authors should give the reader more idea why long-term SDD may not improve the early diagnosis of melanoma by periodical handheld dermoscopy. In conclusion, please focus on the findings in this study and avoid the causal sentence.
Author Response
We thank the reviewer for these comments. This was an exploratory study, and its overarching goal was to assess the performance of SDD and handheld dermoscopy in time. Since we did not anticipate any significant difference between the two techniques, we decided to include all the eligible patients in a 4-year period, without a sample size calculated a priori. We clarified this point in the paragraph on statistics under the patients and methods section. In addition, we considered acceptable for the control group to have twice or three times the sample size of the other group, for a better control of the variability.
We agree that causal sentences should better be avoided here. Following your suggestion, we focused on our findings. The last sentence of the abstract was therefore changed from ”SDD may not have a significant benefit over handheld dermoscopy for the long-term monitoring of low-risk subjects” into “in our long-term monitoring of low-risk subjects, Breslow tumor thickness and pT stage distribution did not differ between handheld periodical dermoscopy and SDD.” The sentence “SDD may not have a significant benefit over handheld dermoscopy for the long-term monitoring of low-risk subjects underlying” was also eliminated from the Conclusions paragraph.
Reviewer 2 Report
Dear Authors
the objective of the paper is to establish the usefulness of SDD for low-risk individuals.
The study is interesting and adding new data in the topic.
Major concerns:
1. Monocentric study. The expertise if the users is influencing the diagnostic accuracy. Were the same MDs evaluating lesions in both arms? Which level of experience?
2. A deeper analysis of demoscopic features of digital dermoscopy subgroup is warranted to understand population characteristics:
- for example, in SDD arm, how many lesions (out if the 59 MMs) were excised because of change of the previously monitored lesion?
- Which type of change was inducing excision?
- How many MMs were "de novo" (not detectable on total body picture)?
- Are there demoscopica differences in patterns / features between digital dermoscopy first time vs. periodical lesions? New 7-point check list score could help to clarify eventual differences between these subgroups
Some of MMs in the SDD group could have been identified during an SDD visit, but not because of a specific digital monitoring of that lesion. In general dermsoscopy practice, at each SDD visit all previous recorded cases are compared but also the other lesions are checked. Also, some changes could have made the MM detectable under simple handheld examination, thus making SDD procedure not relevant for such diagnosis.
These further data may clarify if the reduction in Breslow in this group is related to the SDD procedure or to the frequent patient's examination, or to a different lesion selection and referral.
Author Response
We are grateful to the reviewer for the useful comments.
- This was in fact a multicentric study. At least two different physicians trained in dermoscopy assessed the patients with either handheld dermoscopy or SSD in each center. This was clarified under the patients and methods section.
- The available dermoscopic images have now been studied in all 59 melanoma patients who had undergone SDD and these data are now included in the revised manuscript:
- in the SDD arm, 49 out of 59 lesions (83.1%) were excised because they showed changes since respective previous digital dermoscopy examination.
- Fifteen lesions showed increase in size only, 15 showed change in pattern only and 19 showed both increase in size and pattern change since the latest digital dermoscopy.
- Ten out of 59 lesions (16.9%) were excised because they were not evident on previous total-body photography.
- we found no difference in dermoscopic features and 7-point checklist score between melanomas identified on first-time digital dermoscopy and those detected during SDD.
- we completely agree that a proportion of melanomas identified during SDD could have been detected independently of digital monitoring, because they show dermoscopic features of melanoma since their onset, regardless of change in time. Indeed, there was no significant difference in 7-point checklist score between melanomas detected on SDD, but had not been evident on earlier digital dermoscopic examinations, and those detected on first-time digital dermoscopy. These new data have now been added in the results section. This is also why we hypothesize that different mode of patients referral/selection, probably including socioeconomic and demographic factors, might explain lower Breslow thickness, as stated in the discussion.
Round 2
Reviewer 1 Report
The authors have addressed most of my concerns. The reviewer approves the manuscript for the publication in Cancers in the current state.